# Cardiac Rhabdoid Tumor—A Rare Foe—Case Report and Literature Review

**DOI:** 10.3390/children9070942

**Published:** 2022-06-23

**Authors:** Alina Costina Luca, Ingrith Crenguța Miron, Elena Cojocaru, Elena Țarcă, Alexandrina-Stefania Curpan, Doina Mihăila, Laura Mihaela Trandafir, Alin-Constantin Iordache, Vasile-Valeriu Lupu, Henry D. Tazelaar, Ioana Alexandra Pădureț

**Affiliations:** 1Department of Pediatrics, Faculty of Medicine, “Grigore T. Popa” University of Medicine and Pharmacy, 700115 Iasi, Romania; acluca@yahoo.com (A.C.L.); ingridmiron@hotmail.com (I.C.M.); trandafirlaura@yahoo.com (L.M.T.); valeriulupu@yahoo.com (V.-V.L.); 2Department of Morphofunctional Sciences I—Pathology, “Grigore T. Popa” University of Medicine and Pharmacy, 700115 Iaşi, Romania; ellacojocaru@yahoo.com (E.C.); doinamihaila@umfiasi.ro (D.M.); 3Department of Surgery II, Discipline of Pediatric Surgery, “Grigore T. Popa” University of Medicine and Pharmacy, University Street, no. 16, 700115 Iasi, Romania; 4Department of Biology, Faculty of Biology, University “Alexandru Ioan Cuza”, Carol I Ave, no. 11, 700505 Iasi, Romania; 5Faculty of Medicine, “Grigore T. Popa” University of Medicine and Pharmacy of Iasi, 16 Universitatii Str., 700115 Iasi, Romania; aliniordache@yahoo.com; 6Department of Laboratory Medicine and Pathology, Mayo Clinic Alix College of Medicine, Scottsdale, AZ 85259, USA; tazelaar.henry@mayo.edu; 7Department of Cardiology, Saint Mary Emergency Children Hospital, Vasile Lupu Street, no 62, 700309 Iasi, Romania; paduret.alexandra@gmail.com

**Keywords:** extrarenal rhabdoid tumor, cardiac tumor, case report

## Abstract

Intracardiac masses are unusual findings in infants, and most of them are benign. Nevertheless, they may be associated with a significant degree of hemodynamic instability and/or arrhythmias. Malignant tumors of the heart rarely occur in children. Rhabdoid tumors are aggressive tumors with a dismal prognosis even when diagnosed early. Although rhabdomyomas are common cardiac tumors in infants, they are mostly benign. The most common sites of involvement are the kidneys and central nervous system, but soft tissues, lungs, and ovaries may also be affected. The diagnosis can be challenging, particularly in sites where they do not usually occur. In the present paper, we report the case of a 2-year-old boy diagnosed with cardiac rhabdoid tumor highlighting the importance of molecular studies and recent genetic discoveries with the purpose of improving the management of such cases. The aim of this educational case report and literature review is to raise awareness of cardiac masses in children and to point out diagnostic hints toward a cardiac tumor on various imaging modalities. Given the rarity of all tumors involving the heart and the lack of symptom specificity, a high degree of suspicion is needed to arrive at the correct diagnosis.

## 1. Introduction

Being rare, cardiac tumors pose real diagnostic and management challenges. The estimated incidence of primary cardiac tumors is 1:100,000 people, but autopsy reports establish the prevalence of primary neoplasms at 1:2000, while metastatic tumors have an estimated prevalence of 1:100 [1]. Recent advances in cardiac imaging, leading to better tissue characterization of intracardiac masses, have improved our ability to diagnose and treat them [2].

Herein, we report the case of a 2-year-old boy diagnosed with a cardiac rhabdoid tumor. The aim of this educational case report and literature review is to raise awareness of cardiac masses in children and to point out diagnostic markers for a cardiac tumor on chest X-ray, echocardiography, and chest computerized tomography (CT). Moreover, the vital role of immunohistochemistry tools and genetic investigation is highlighted and facilitates the differential diagnosis of a cardiac mass in children.

### 1.1. Benign Cardiac Tumors in Infants and Children

Myxomas, rhabdomyomas, and fibromas are the most frequently diagnosed tumors in infancy and childhood. Rhabdomyomas are the main type of primary cardiac tumors, accounting for 80% of cases [3]. In infants and children, they are usually associated with tuberous sclerosis, vacuolated cells with eccentric nuclei, and sparse cytoplasm [4,5]. 

Myxomas account for 6% of diagnosed tumors in children [3,6]. Patients may develop symptoms similar to mitral or tricuspid valve stenosis due to the frequent obliteration of the valve lumen by a mobile myxoma, arising either in the right or left atrium [7]. Myxomas can occur in a familial context due to a mutation in the *PRKAR1A* gene. In this setting, they are associated with ephelides, mucocutaneous myxomas, lentigines, or naevi, a constellation known as the Carney complex, an autosomal dominant syndrome [8].

Fibromas are congenital cardiac masses associated with Gorlin or Gardner syndrome [3,4]. They usually have large dimensions and present as intramural, solitary masses located in the interventricular septum, the left ventricular free wall, or apex. Histologically, fibromas are rich in fibroblasts and collagen fibers and present with a scarcity of elastic fibers [3,7]. They are usually associated with ventricular arrhythmias and cardiac arrest [9].

### 1.2. Malignant Tumors of the Heart in Infants and Children—Extrarenal Rhabdoid Tumor

Rhabdoid tumors are rare and aggressive neoplasms developing mainly in the kidney but also in cerebral and extra-cerebral tissues [10] and rarely in the heart. Diagnosis is challenging, but immunohistochemical evaluation allowing detection of *INI-1* loss has made the diagnosis of this entity easier. INI-1 loss reflects the presence of *SMARCB1* mutations, a sensitive marker for malignant rhabdoid tumors [11].

Rhabdoid tumors can occur in the setting of a more complex clinical entity now known as rhabdoid tumor predisposition syndrome (RTPS). Such a diagnosis should be suspected if a patient presents with any of the following: 1. Atypical teratoid/rhabdoid tumor (rhabdoid tumor affecting the central nervous system); 2. Rhabdoid tumor of the kidney; 3. Rhabdoid tumors of the heart, liver, mediastinum, retroperitoneum, bladder, and pelvis; 4. Small cell carcinoma, the hypercalcemic type of the ovary [12]. 

Histologically, rhabdoid renal and extrarenal tumors share common characteristics. The cells are polygonal, with eccentric, vesicular nuclei and prominent nucleoli, eosinophilic cytoplasmic inclusions [13]. The tumors have an infiltrative growth pattern, necrosis, and high proliferative index [14]. The immunohistochemistry tools of investigation facilitate differential diagnosis by showing loss of *INI-1* expression, usually accompanied by germline mutations of the *SMARCB1* gene. When *INI-1* is still expressed in the affected tissues, a rhabdoid tumor with *SMARCA4* mutations should be considered [14]. 

The incidence of atypical teratoid/rhabdoid tumor (AT/RT) in children younger than 1 year is estimated at 5.4:10 [15]. Carriers of the *SMARCB1* mutation have the RTPS 1 type (OMIM #609322), and those carrying the *SMARCA4* mutation are diagnosed with type 2 RTPS (OMIM #613325). The inheritance pattern is autosomal dominant, although the penetrance is yet to be established. Additional loss of function or missense mutations as a second hit phenomenon have been involved in the occurrence of different types of syndromes associated with *SMARCB1* and *SMARCA4* germline mutations [16].

Given that the age of symptom onset is approximately 2 years old, and the 5-year survival rate is 10%, the surveillance of the proband and familial studies are paramount. The surveillance guidelines suggest physical examination every 2–3 months and imaging studies with a frequency dictated by age (Table 1) [16,17].

According to the WHO Histological Classification of Tumors of the Heart and Pericardium, the majority of malignant primary tumors of the heart can be categorized as various types of sarcomas, with primary cardiac lymphoma and epithelioid hemangioendothelioma accounting for a small fraction of cases. Primary malignant tumors are very rare in infants and children. Rhabdomyosarcoma and teratoma are the most frequent culprits (Table 2) [18]. 

### 1.3. Diagnosis and Classification

Transthoracic echocardiography (TTE) is a valuable method for determining the location, morphology, mobility, and density of a cardiac mass. The hemodynamic impact of a tumor can be evaluated by means of continuous and color Doppler, while speckle tracking helps in establishing the masses’ contractility [7,26]. 

Transesophageal echocardiography is especially useful in evaluating atrial tumors and renders superior results to cardiac magnetic resonance imaging in valvular masses [26,27]. 

Chest X-ray is a cost-effective and widely used imaging tool for dyspnea, cough, and chest pain. It allows for the detection of an increase in heart size but is non-specific and can also be observed in chronic heart failure or pericardial effusion.

Cardiac computed tomography (CT) and magnetic resonance (CMR) have proven useful for preoperative evaluation. CT detects calcifications and cardiac valve masses with a superior accuracy compared to CMR, which makes it the favored imagistic tool for planning reconstruction. When a differential diagnostic is necessary, CMR is indicated, especially in pediatric cases [15,28,29].

Positron emission tomography is used when differentiating between the malignant and benign nature of a tumor is necessary. Increased metabolic activity is a pathognomonic sign of a neoplasm, although false-positive results may be identified in inflammatory and infectious conditions [30,31]. The main imaging modalities and differential diagnoses for cardiac tumors are listed below in Table 2.

## 2. Material and Methods

This is a case report of one patient diagnosed with a cardiac rhabdoid tumor admitted to the Pediatric Cardiology Department of ‘St. Maria’ Emergency Children’s Hospital of Iași. The present study was conducted according to Romanian research law no. 206/27.05.2004 as well as the European laws. The parents were informed about the study, what was involved, and what information was going to be used, and approval from the Ethics Committee of “Saint Mary” Emergency Children’s Hospital was also obtained.

For the literature review, the Medical Subject Headings MeSH terms extrarenal rhabdoid tumor and cardiac tumor were used in PubMed searching for randomized controlled trials (RCTs), systematic reviews, observational studies, case series, and case reports from the earliest possible date to February 2021, published in English. Additional articles were identified in the references of the aforementioned papers.

## 3. Results

The patient was a 2-year-old male with an 8 APGAR birth score. He exhibited normal development and was admitted due to productive cough, dynamism, drowsiness alternating with psycho-motor agitation, fever, and bilateral seromucous ear secretions. The patient had multiple previous hospital admissions for cases of pneumonia conditions due to a previous tuberculosis exposure, diagnosed through clinical examination, paraclinical investigations, and radiology.

Upon clinical examination, he presented with a weight deficit (weight = 9 SD, height = 26 SD), pale skin, serous ear secretion, bilateral lungs vesicular murmur, a heart rate of 118 beats per minute, and peripheral oxygen saturation of 92%. Chest radiography (Figure 1 and Figure 2) showed left lung opacities with blurred edges, an unaffected left lung tip, and left costodiaphragmatic recess. Another opacity that occupied the anterior mediastinum was also identified.

A gastric lavage sample and blood culture were negative for Koch Bacillus testing.

Abdominal ultrasound revealed a liquid blade in the peritoneal cavity with a thickness of 1.2 cm. After a soft tissue ultrasound, he was found to have right lateral cervical adenopathy of 37/19 mm, positioned 5 mm subcutaneously, posterior to the sternocleidomastoid (SCM) muscle, in the lower cervical floor, with apparently present Doppler signal.

Despite multiple courses of antibiotics, his condition worsened. A contrast thoracic computer tomography (CT) (Figure 3 and Figure 4) was performed. The results were very concerning, as a tumor mass of 8.57 × 10.37 × 9.42 cm (anteroposterior, AP × transversal, T × craniocaudal, CC) with native soft tissue densities (18–30 UH) was identified. It was moderately iodophilic, non-homogeneous, had a straight polycyclic contour, located in the upper and middle anterior mediastinum, fully occupying the retrosternal space with a prominent left paramedian extension. The tumor exerted a mass effect on the left main bronchus, reducing its caliber up to the lobular bifurcation, surrounded two-thirds of the right intermediate bronchus posteriorly, and encompassed the trunk of the pulmonary artery, the right and left branch of the pulmonary artery, the ascending aorta, the aortic cross and the branches emerging from it. The vascular lumens were preserved. There was also an adenopathy located latero-cervically inferior to the right, 1.5 × 0.9 × 1.86 cm (AP × T × CC), located posterior to the right sternocleidomastoid muscle, compressing it and displacing it anteriorly. A pericardial effusion with variable thickness between 1.46 cm and 2.11 cm and axillary lymph node localized 1-cm-in-diameter could also be seen.

Based on the initial CT, there was suspicion of lymphoma, but additional workup failed to confirm this possibility. Flow cytometry revealed no atypical lymphocytes in the pleural fluid sample. The malignant hematological disease monitoring test identified large, mature monocytes with the following phenotypes: CD45 HLA/DR+; CD34-; CD117-; CD64+; CD36+; CD14+; IREM-2+. Pleural fluid cytological examination described numerous mesothelial cells and monocyte-macrophages, lymphocytes, and polymorphonuclear cells in approximately equal proportions. Extended investigations revealed non-infiltrated bone marrow through medullary puncture and sinus histiocytosis at nodal biopsy.

Biologically, the patient presented leukocytosis with neutrophilia, moderate anemia, low fibrinogen levels, significantly increased C-reactive protein, high D-dimers levels, severe hypoproteinemia, hyponatremia, and metabolic acidosis.

After two weeks of medical management with no improvement, he developed upper body edema, oxygen desaturation, and psycho-motor agitation, and so he was transferred to the intensive care unit. The CT was repeated and showed that the tumor had evolved. It had grown to 8.22/11.30/110 cm (AP/T/CC) and now exhibited inhomogeneity, with areas of necrosis and contrast settings at the level of solid components. The tumor now encompassed the left common carotid arteries, left subclavian artery, right brachiocephalic arterial trunk, pulmonary artery trunk, right and left branches of the pulmonary artery, and the superior vena cava (SVC) with a non-occlusive parietal thrombus in the middle to the inferior third of the SVC (0.81/0.73 cm), which developed into complete obstruction of the left brachiocephalic venous trunk. The left ventricular wall showed non-homogeneous contrast. There was also an adenopathy located at the high mediastinum with dimensions of 2.03 × 1.43 cm.

The patient’s condition deteriorated, and he developed acute liver failure and superior vena cava syndrome. He was intubated and mechanically ventilated. Twenty-seven days after hospitalization, an irreversible cardio-respiratory arrest occurred, and the patient died. An autopsy was performed.

The gross evaluation (Figure 5) revealed the presence of a cardiac tumor that infiltrated the pericardium with lung metastases, pulmonary congestion, mediastinal lymphadenopathy, passive hepatic congestion and massive steatosis, extensive thrombosis of the superior vena cava, left brachiocephalic vein, and left internal jugular vein thrombosis. The cardiac tumor appeared to arise at the level of the atrial wall and involved almost the entire heart. When sectioned, the remaining myocardium was found only at the tip of the heart.

Sections of the cardiac tumor revealed the proliferation of large tumor cells with marked cyto-nuclear pleomorphism, atypical mitosis (Figure 6), necrosis area, haemorrhage, and vascular tumor embolism. Immunohistochemical studies were performed on paraffin-embedded tissue for tumor cell phenotyping. The cells were reactive with antibodies to cytokeratin AE1/AE3 (Figure 7) and S100 protein, but they failed to react with antibodies to SMA (Figure 8), vimentin (Figure 9), myogenin, calretinin, and melan-A. This made us consider the diagnoses of an epithelioid sarcoma or myoepithelial carcinoma, but additional immunohistochemical and molecular biology tests were recommended for a definite diagnosis. Thus, tissue fragments were sent to another pathology department where INI 1 expression was studied, and the tumor process showed a loss of expression of this protein (Figure 10). This led to the definite diagnosis of extrarenal malignant cardiac rhabdoid tumor.

## 4. Discussion

Due to the very few cases available for studies, randomized controlled trials for rhabdoid tumors in children have not been developed. According to the EU-RHAB registry, total resection, standard chemotherapy, intrathecal methotrexate (MTX), high-dose chemotherapy with Carboplatin and Thiotepa, and radiotherapy are recommended in children over 18 months of age. This regimen is associated with significant survival rate improvement, but it is associated with numerous adverse side effects [32]. Rarely, such therapy might be considered in younger patients. Multiple phase I and II studies are currently underway evaluating a series of inhibitors targeting histone deacetylase, histone methyltransferase DNA methyltransferase, Aurora kinase A, and the Hedgehog pathway in order to establish new therapeutic regimens that may help decrease the chemotherapy dosage and improve prognosis [33]. In general, the clinical course of malignant cardiac tumors is characterized by aggressive growth and fatal outcome. Patients with primary malignant diseases or metastases may undergo surgery for symptomatic and palliative considerations.

Our patient presented with lung disease and was unresponsive to multiple antibiotic treatments, which in the end were found to be due to massive involvement of the thoracic organs by cardiac rhabdoid tumors [13]. This type of neoplasm is slightly more encountered in the male population, which was also the case with our patient [34].

The main differential for the tumor in our case was an epithelioid sarcoma. Both tumors can have rhabdoid morphology, with a loss of *INI-1*, while epithelial and mesenchymal markers are positive, whereas rhabdoid tumors express epithelial, mesenchymal, and neural markers [11], and both tumors show immunopositivity for keratins and EMA, with occasional staining for desmin and CEA. CD34 can be used as a differential tool since epithelioid sarcoma is CD34 positive in 50% of cases, while rhabdoid tumors are always negative [35].

The identification of such a young patient with a malignant rhabdoid tumor meets the criteria necessary to suspect a predisposition syndrome. Given the rarity of a rhabdoid tumor being malignant, since they are commonly benign, ideally, a family investigation should be conducted in order to identify all cases with a diagnosis of rhabdoid tumor and/or multiple germline mutations of *SMARCB1* or *SMARCA4* through molecular testing [12]. These tumors may be metachronous or synchronous. Both *SMARCB1* and *SMARCA4* gene mutations have been implicated in Coffin-Siris syndrome and *SMARCB1* for schwannomatosis, an important aspect to remember when conducting family clinical investigation [12]. *SMARCA4* mutations are also linked to lung cancer [36]. It seems rational to test for *SMARCB1* mutations first since the vast majority of rhabdoid tumor predisposition syndrome cases are caused by variants of this gene, and those cases have a more reserved prognosis, hampered by the recurrence of tumors, their synchronous presence in different sites, and the high probability of developing masses in the central nervous system if it was not initially affected.

The treatment of a rhabdoid tumor is multimodal. Good results were obtained with induction therapy by means of Cyclophosphamide, Cisplatin, Etoposide, Methotrexate, and Vincristine, followed by consolidation therapy with Carboplatin and Thiotepa [37]. Radical surgery has been suggested as the best therapeutic option; however, the tumor size and location seldom allow for such interventions. The European Rhabdoid Registry recommends high-dose chemotherapy, Methotrexate, and radiotherapy [38].

Unfortunately, in the case of our patient, his definitive diagnostis came too late, significantly lowering the already slim chances of survival.

In any case, provided that a prompt and early diagnosis is made and the patient is responsive to the multimodal treatment scheme, long-term surveillance will be necessary. The guidelines also stress the importance of genotyping, as *SMARCB1* mutations are associated with a higher risk for abdominal neoplasm, which is why an MRI every 5 years and an ultrasound every 3 months are appropriate. *SMARCA4* mutations, if identified, are more likely to cause small cell carcinoma of the ovary, thus prompting abdominal ultrasound every 6 months to be a wise course of action [16].

## 5. Conclusions

Rhabdoid malignant tumors (RMT) are exceptionally aggressive neoplasms, though rare. The old paradigm of kidney and CNS involvement is now obsolete, as genetic studies make it clear these tumors occur in various other locations, including thoracic locations. Rhabdoid predisposition syndrome is caused by a germline mutation in either *SMARCB4* or *SMARCA1* genes and should always be considered when dealing with a patient diagnosed with an RMT. The only way to increase even slightly the chances of survival is by fast recognition and prompt positive diagnosis facilitated by immunohistochemical and molecular studies. Management is individualized as there are no accepted guidelines, but it must be aggressive, prompt, and multimodal. Newer targeted therapies are currently under development in order to minimize the long-term associated side effects.

## Figures and Tables

**Figure 1 children-09-00942-f001:**
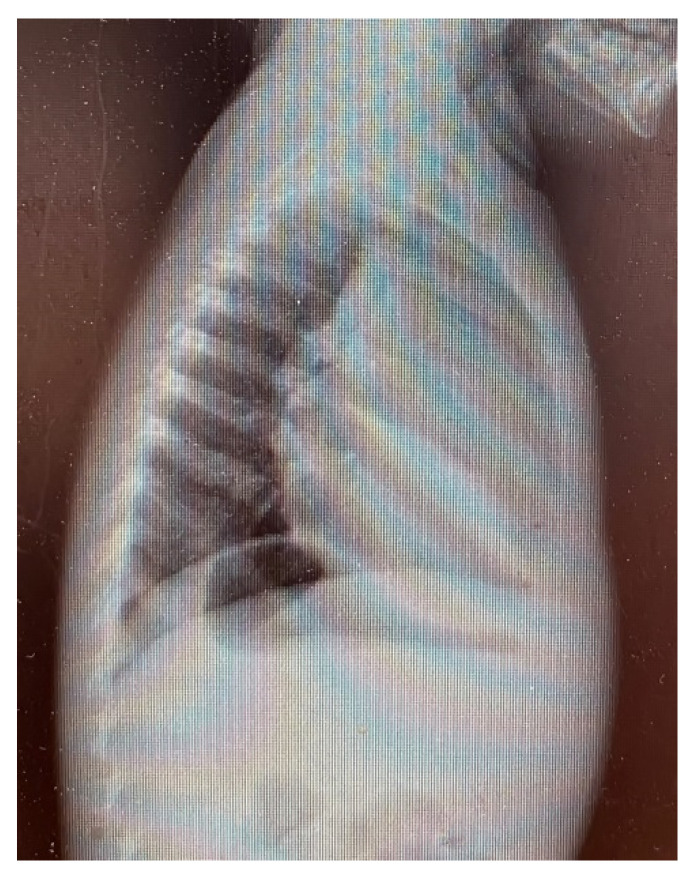
Lateral chest radiography.

**Figure 2 children-09-00942-f002:**
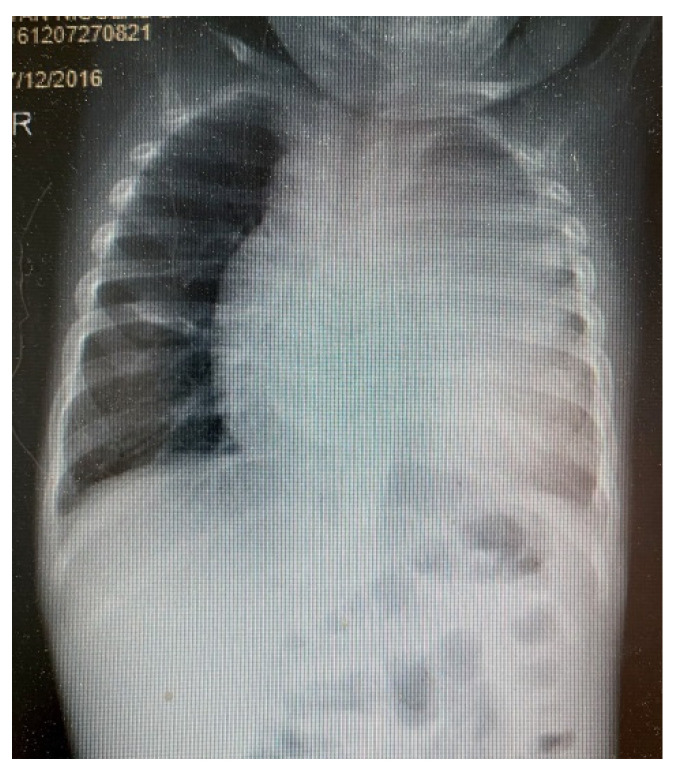
Anteroposterior chest radiography.

**Figure 3 children-09-00942-f003:**
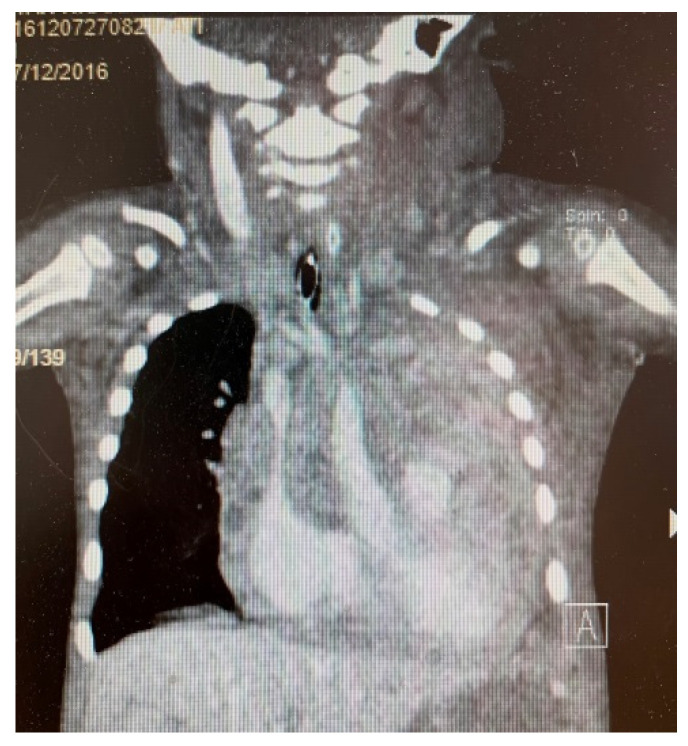
CT: a tumor mass located in the upper and middle anterior mediastinum, fully occupying the retrosternal space with prominent left paramedian extension.

**Figure 4 children-09-00942-f004:**
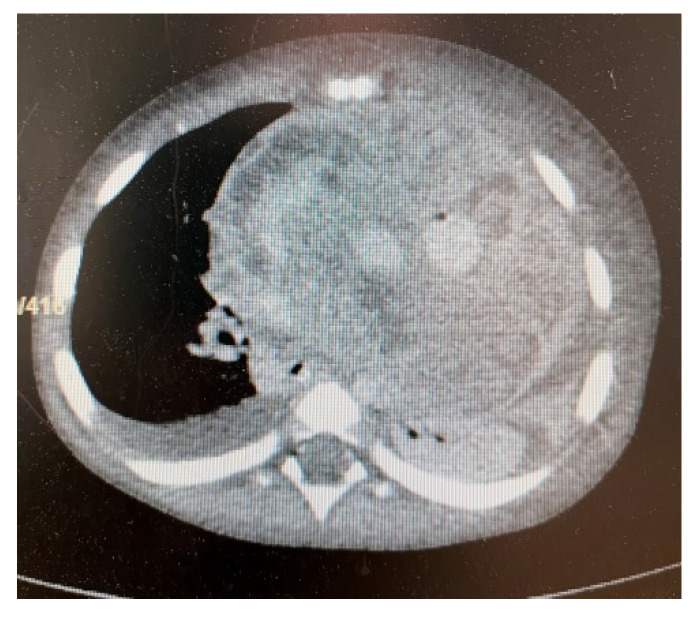
CT aspect of the tumor.

**Figure 5 children-09-00942-f005:**
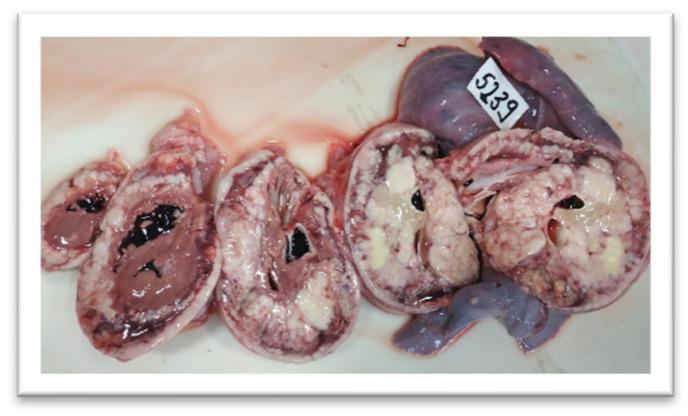
Malignant cardiac tumor developed in the heart with invasion of the entire pericardium.

**Figure 6 children-09-00942-f006:**
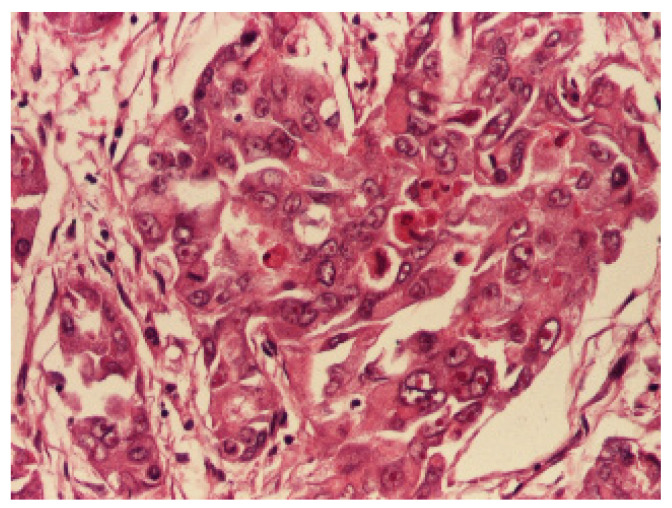
Large tumor cells with marked cyto-nuclear pleomorphism and atypical mitosis, HEx 200.

**Figure 7 children-09-00942-f007:**
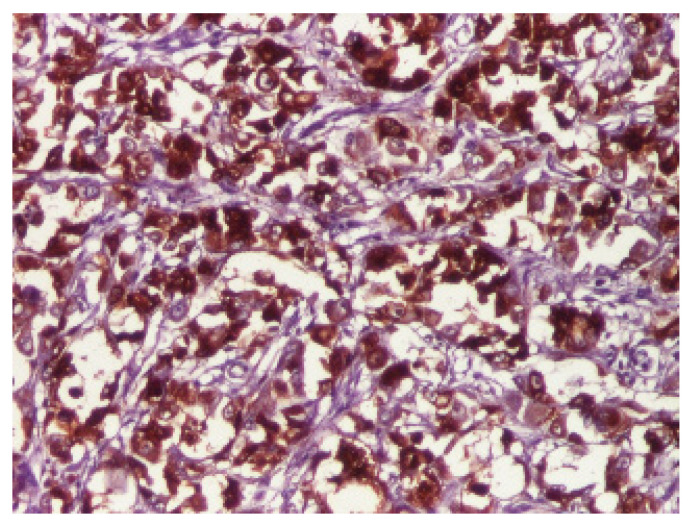
Immunohistochemical expression of AE1_AE3 showed a diffuse strong staining in tumor cells, ×100.

**Figure 8 children-09-00942-f008:**
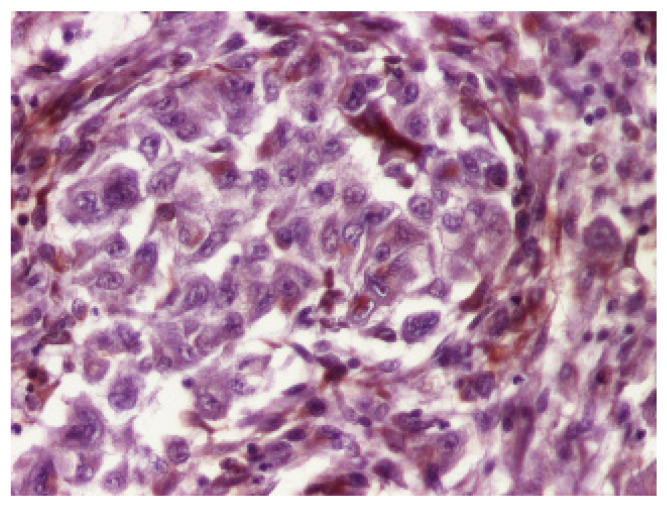
The tumor cells showed no expression of vimentin, ×200.

**Figure 9 children-09-00942-f009:**
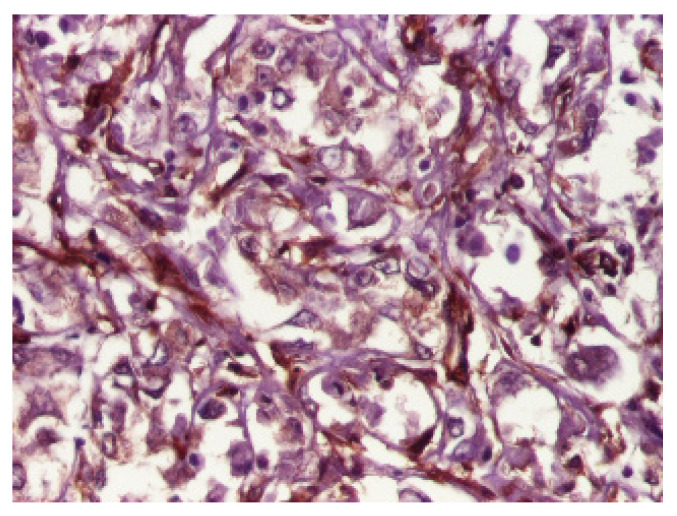
SMA was not expressed by the tumor cells, ×200.

**Figure 10 children-09-00942-f010:**
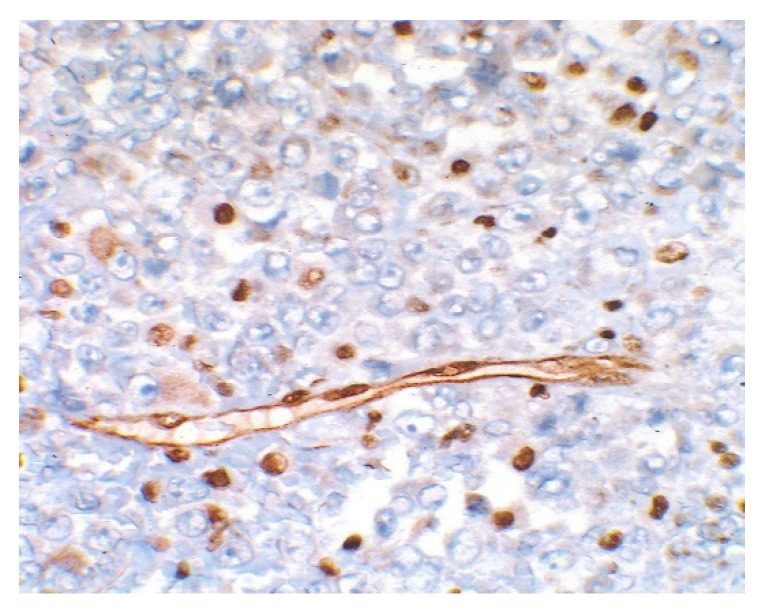
Loss of INI 1 immunohistochemical expression in tumor cells, ×1004.

**Table 1 children-09-00942-t001:** Frequency of MRI and/or ultrasound examination in RTPS 1 and RTPS 2 based on the age of the patient.

Age	Imagistic Studies	Frequency
All	Whole-body MRI	After *SMARCAB1* mutation discovered
0–6 months	Whole-body MRI or CNS MRI Abdomen and soft tissue ultrasound	Every 4 weeks, not less than every 2–3 months
7–18 months	Whole-body MRI or CNS MRI Abdomen and soft tissue ultrasound	Every 2–3 months
19 months–5 years	Whole-body MRI or CNS MRI Abdomen and soft tissue ultrasound	Every 3 months
>5 years	Whole-body MRI	Yearly

**Table 2 children-09-00942-t002:** Differential diagnosis for myxoma, fibroma, rhabdomyoma, and rhabdomyosarcoma with their clinical manifestations and imagistic criteria based on *ECG-electrocardiogram, *TTE/TEE-Transthoracic echocardiography, *CT-Cardiac computed-tomography, *CMR-cardiac magnetic resonance, biomarkers as well as therapeutic options.

Tumor Type	Clinical Manifestations [19]	*ECG [20,21,22]	*TTE/ TEE [23]	*CT [20,23]	*CMR [20,23]	*Biomarkers [24]	Differential Diagnosis [24]	Therapy [20,24,25]
Myxoma	Flow-Obstruction; Emboli; Systemic symptoms;	Left atrial enlargement; Ventricular tachycardia;	Narrow stalk; Hyperechoic mass in characteristic location; Calcifications; Dynamic tumor;	Low-attenuation heterogeneous mass compared with myocardium; Pulmonary infarction; Intratumoral calcification;	T1 hypointense, T2 hyperintense Heterogeneously enhancing isointense or hyperintense on delayed imaging;	CD31 + CD34+ Calretinin + CD68-Cytokeratins-	Left atrial thrombus; Metastatic carcinoma; Myxoid sarcoma; Papillary fibroelastoma; Fibroma;	Surgical excision
Fibroma	Heart murmurs; Congestive heart failure; Arrhythmias; Sudden death;	T-wave abnormalities; Ventricular tachycardia; Atrioventricular block;	Large, solid, heterogeneous mass that is noncontractile	Central calcification within a discrete mass; Non-specific low attenuation mass;	Encapsulated mass; Delayed enhancement; Hypointense to isointense in T1; Hypointense in T2;	Vimentin + Ki-67- CD34- S100- HMB45-	Cardiac rhabdomyoma; Myxoma; Teratoma; Lipoma; Hemangioma; Hypertrophic cardiomyopathy; Metastatic disease	Amiodarone and/or beta-blockers; Surgical excision; Single ventricle palliation; Cardiac transplant;
Rhabdomyoma	Flow obstruction; Heart failure; Arrhythmias; Decreased peripheral pulses and/or cyanosis	Extrasystoles; Ventricular Tachycardia; Supraventricular tachycardia; Wolff–Parkinson–White syndrome	Solid, hyperechoic, avascular mass; Focal abnormality of cardiac wall motion	Hypodense compared with adjacent myocardium	T1 isointense/slightly hyperintense; T2 hyperintense; No fat suppression	Myoglobin + Actin + Desmin + Vimentin + S100-	Glycogen storage disease; Granular cell tumor; Lipoma;	mTOR inhibitors; Surgical excision if located in the left ventricle;
Rhabdomyosarcoma	Systemic illness; Syncope; Arrhythmias; Sudden death; Pericardial disease or tamponade; Embolic phenomena	Ventricular arrhythmias	Solid, hyperechoic mass with irregular borders	Hypoattenuating mass involving any cardiac chamber; Smooth or irregular borders	Heterogeneous mass with high signal intensity in T2	Myogenin+ MSA + MYOD1 + Desmin+	Angiosarcoma; Fibrosarcoma; Osteosarcoma; Leiomyosarcoma; Liposarcoma; Lymphoma; Intrapericardial pheochromocytoma; Metastatic disease	Surgical resection; Heart transplantation; USA chemotherapy: Vincristine + Actinomycin-D + Cyclophosphamide; EU chemotherapy: Ifosfamide + Vincristine + Actinomycin-D

## Data Availability

All data is included in the present paper.

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
