# Peer review of "Cardiac Rhabdoid Tumor—A Rare Foe—Case Report and Literature Review"

_children, 2022, doi:10.3390/children9070942_

Round 1
Reviewer 1 Report
Thank you for allowing me to review "Cardiac Rhabdoid Tumor - a Rare Foe – Case Report and Literature Review"
My comments:
remove last sentence of abstract
weight deficit (weight = 9 SD, height = 26 SD) - I am not able to understand this data
not all rhabdoid tumors are malignant sarcomas. In fact rhabdomyomas are common cardiac tumors in infants and they are mostly benign. Please change the wording to delineate this difference in discussion and abstract
Author Response
Dear Reviewer 1,
Thank you for your comments as they are valuable observations and we decided to answer as follows:
- remove last sentence of abstract
After careful consideration, we have come up to the conclusion that indeed that last sentence wasn't the best used in the abstract and didn't quite fit therefore we have removed it.
- weight deficit (weight = 9 SD, height = 26 SD) - I am not able to understand this data
This refers to the standard deviation from what value it should have.
- not all rhabdoid tumors are malignant sarcomas. In fact rhabdomyomas are common cardiac tumors in infants and they are mostly benign. Please change the wording to delineate this difference in discussion and abstract
We are grateful for this observation as we might have omitted this or not made it clear enough throughout the manuscript therefore we have highlighted it both in the discussion and abstract.
Thank you for your time and valuable observations
Reviewer 2 Report
This was an interesting case report with a lot of good, supportive information. One thing to consider is that there is a lot of background information about the various benign and malignant tumors currently in the results section that may be better placed in the introduction (or even possibly discussion) sections.
Author Response
Dear reviewer 2,
Thank you for your kind words and we have carefully considered your request:
"One thing to consider is that there is a lot of background information about the various benign and malignant tumors currently in the results section that may be better placed in the introduction (or even possibly discussion) sections".
We commonly agreed that we should split up the background information from the results to the introduction and discussion sections. We hope this is a more suitable and coherent organization of the paper.
Thank you for your time and suggestion.
Reviewer 3 Report
A paper of great educational importance in terms of the clinical presentation of a rare tumor. Immunohistochemical and molecular diagnostic possibilities are presented. The significance of SMARCA/B mutations, especially in the aspect of predisposition to neoplasia, was emphasized, which is important in family monitoring. A comprehensive differential diagnosis of malignant and benign cardiac tumors was performed. This paper is of great practical importance.
Author Response
Dear Reviewer 3,
Thank you for your time and comments. We are pleased to know you find our paper to be of importance as well as interesting.